

# Characterization of bidirectional gene pairs in The Cancer Genome Atlas (TCGA) dataset

Juchuanli Tu[1,2], Xiaolu Li[2] and Jianjun Wang[3]

[1] College of Life Sciences, Beijing Normal University, Beijing, China
[2] National Institute of Biological Sciences, Beijing, China
[3] State Key Laboratory of Microbial Resources, Institute of Microbiology, Chinese Academy of Sciences, Beijing, China

Corresponding author
Jianjun Wang, wangjj@im.ac.cn

## ABSTRACT

The "bidirectional gene pair" indicates a particular head-to-head gene organization in which transcription start sites of two genes are located on opposite strands of genomic DNA within a region of one kb. Despite bidirectional gene pairs are well characterized, little is known about their expression profiles and regulation features in tumorigenesis. We used RNA-seq data from The Cancer Genome Atlas (TCGA) dataset for a systematic analysis of the expression profiles of bidirectional gene pairs in 13 cancer datasets. Gene pairs on the opposite strand with transcription end site distance within one kb or on the same strand with the distance of two genes between 1–10 kb and gene pairs comprising two randomly chosen genes were used as control gene pairs (CG1, CG2, and random). We identified and characterized up-/down-regulated genes by comparing the expression level between tumors and adjacent normal tissues in 13 TCGA datasets. There were no consistently significant difference in the percentage of up-/down-regulated genes between bidirectional and control/random genes in most of TCGA datasets. However, the percentage of bidirectional gene pairs comprising two up- or two down-regulated genes was significantly higher than gene pairs from CG1/2 in 12/11 analyzed TCGA datasets and the random gene pairs in all 13 TCGA datasets. Then we identified the methylation correlated bidirectional genes to explore the regulatory mechanism of bidirectional genes. Like the differentially expressed gene pairs, the bidirectional genes in a pair were significantly prone to be both hypo- or hyper-methylation correlated genes in 12/13 TCGA datasets when comparing to the CG2/random gene pairs despite no significant difference between the percentages of hypo-/hyper-methylation correlated genes in bidirectional and CG2/random genes in most of TCGA datasets. Finally, we explored the correlation between bidirectional genes and patient's survival, identifying prognostic bidirectional genes and prognostic bidirectional gene pairs in each TCGA dataset. Remarkably, we found a group of prognostic bidirectional gene pairs in which the combination of two protein coding genes with different expression level correlated with different survival prognosis in survival analysis for OS. The percentage of these gene pairs in bidirectional gene pair were significantly higher than the gene pairs in controls in COAD datasets and lower in none of 13 TCGA datasets.

## INTRODUCTION

The "bidirectional gene pair" indicates a particular "head-to-head" gene organization in which transcription start sites (TSSs) of two genes are located on opposite strands of genomic DNA within a region of one kb (*Lin et al., 2007*; *Liu, Chen & Shen, 2011*; *Trinklein et al., 2004*; *Yang, Koehly & Elnitski, 2007*). The intervening regulatory region is called "bidirectional promoter" (*Liu, Chen & Shen, 2011*). Bidirectional genes have been thought to represent more than 10% of human genes (*Yang, Koehly & Elnitski, 2007*). The architecture of bidirectional gene pairs is evolutionarily conserved, thus suggesting a functional importance (*Koyanagi et al., 2005*; *Trinklein et al., 2004*; *Yang & Elnitski, 2014*). Indeed, some of bidirectional genes are often associated with functions in DNA repair, with the potential to participate in the development of cancer (*Adachi & Lieber, 2002*; *Koyanagi et al., 2005*; *Yang, Koehly & Elnitski, 2007*). Previous studies indicated that bidirectional gene pairs are associated with several genomic regulatory features such as CpG island methylation and histone modification (*Adachi & Lieber, 2002*; *Jangid et al., 2018*; *Lin et al., 2007*; *Shu et al., 2006*; *Trinklein et al., 2004*).

As previously indicated, bidirectional gene pairs are related to crucial cellular functions and contribute to tumorigenesis (*Adachi & Lieber, 2002*; *Yang, Koehly & Elnitski, 2007*). However, the majority of these studies are based on genomic features without considering the real expression level of bidirectional gene pairs in cancer. Besides, little is known about the regulation mechanisms underlying bidirectional gene pairs involved in tumorigenesis.

Here, we used RNA-seq data from The Cancer Genome Atlas (TCGA) dataset for a systematic analysis of the expression profiles of bidirectional genes and gene pairs in cancer (*Cancer Genome Atlas Network, 2012a*, *2012b*; *Cancer Genome Atlas Research Network, 2012*, *2013*, *2014a*, *2014b*). We identified and characterized the up-/down-regulated bidirectional genes and gene pairs in 13 TCGA datasets. We also identified and characterized the methylation correlated bidirectional genes to explore the regulatory mechanism of bidirectional genes. Besides, we evaluated the prognosis of bidirectional genes by survival analysis to correlate the expression of bidirectional genes with clinical outcome.

## MATERIALS AND METHODS

### Identification of bidirectional, control, and random gene pairs in the human genome

The gene annotation GDC.h38 v.22 was retrieved from GENCODE (https://gdc.cancer.gov/about-data/data-harmonization-and-generation/gdc-reference-files). The definition of gene type for each gene was included in the annotation file.

For identifying bidirectional gene pairs, we used a method proposed in a previous work (*Trinklein et al., 2004*). Briefly, for each gene we defined the position of the TSS and the transcription end site (TES) as the 5′-most boundary and the 3′-most boundary transcripts of the analyzed gene, respectively (head-to-head structure). We defined bidirectional genes as two genes on opposite strands with TSS distance within one kb. We excluded that gene pairs in which a gene was entirely located in another gene, a condition commonly defined as nested genes.

For identifying two control gene pairs (CG1 and CG2), we followed the same way except for searching the gene pairs on the opposite strand with TES distance within one kb as control gene pairs 1 (CG1) and on the same strand with the distance of two genes between one and 10 kb as control gene pairs 2 (CG2) (*Trinklein et al., 2004*). Finally, we excluded control gene pairs which contain bidirectional gene.

Since the number and the composition of gene types for bidirectional gene pairs and control gene pairs above was the difference, we also collected a set of gene pairs comprising two genes randomly chosen in the genome as control. The number and the composition of gene types for each set of random gene pairs was the same as bidirectional gene pairs. We repeated 100 times of selections and got 100 sets of random gene pairs.

## TCGA expression datasets and analysis

All TCGA expression datasets from RNA-seq were downloaded from the TCGA website (https://portal.gdc.cancer.gov/). Raw reads counts were extracted from files with the suffix "htseq.counts." We only incorporated cancer datasets with ≥200 cancer samples and ≥15 normal samples for an accurate identification of differentially expressed genes. The normal samples were defined as the samples marked as "Solid Tissue Normal" and no blood samples were included into the analysis as normal samples. A total of 13 cancer datasets matched these criteria. All subsequent analyses were performed on these 13 cancer datasets except for the methylation data analysis. We excluded the STAD dataset in methylation analysis for the methylation data available for only two normal samples in this dataset.

We employed the "RUVg" function in "RUVSeq" package (v3.8) to correct the batch effect in RNA-seq datasets following the previous studies (*Aran et al., 2017*; *Risso et al., 2014*). The identification of differentially expressed genes was performed by "glmTreat" function in "edgeR" package (v3.24.0) (*Robinson, McCarthy & Smyth, 2010*). We kept genes in each TCGA dataset when their count-per-million was ≥0.1 in at least the size of normal samples and defined these genes as expressed genes. Then we identified the differentially expressed genes by comparing the expression profile of a specific gene between cancer and normal sample groups. Genes with FDR ≤0.05 and expression fold change ≥1.5 (up-regulated genes) or ≤0.67 (down-regulated genes) were defined as differentially expressed genes (*Xiao et al., 2018*). The trimmed mean of M-values (TMM) normalized expression value was also generated by "edgeR" package.

## Gene ontology analysis

The gene ontology (GO) analysis was performed on the DAVID GO bioinformatics platform (V6.8) (https://david.ncifcrf.gov/) (*Dennis et al., 2003*). The ensembl gene ID for each gene was used as the input. Genes which were not recognized by DAVID GO website were excluded from GO analysis.

## Correlation analysis of methylation level and bidirectional genes and gene pairs

The DNA methylation data from Human Methylation 450 BeadChip platform were downloaded from TCGA website (https://portal.gdc.cancer.gov/). We extracted beta-values

to evaluate the DNA methylation level of each probe. The annotations of probes to specific genes were defined as these probes were located on the promoter region of genes. The promoter region was defined as the region from two kb upstream to 500 bp downstream of TSS of a gene (*Tang & Epstein, 2007*). We excluded the STAD dataset for the methylation data available for only two normal samples.

We used the "champ.DMP" function in the "ChAMP" package in R to identify differentially methylation probes (*Morris et al., 2014*). We defined probes with adjusted $p$-value $\leq 0.05$ as differentially methylation probes.

Then we defined a gene as correlated to methylation when this gene was up-regulated in cancer with at least one hypo-methylated probe was annotated to this gene, or when this gene was down-regulated in cancer with at least one hyper-methylated probe was annotated to this gene.

## Survival analysis in TCGA datasets

The survival analysis was basically performed as previously reported (*Anaya et al., 2015*). Briefly, for each cancer dataset, survival information were retrieved from the "TCGA-CDR" dataset, which keeps for each patient the most recent follow-up information (*Liu et al., 2018*). Since the information of disease-specific survival is approximated in most of TCGA datasets, we excluded it from our analysis and kept other three endpoints (overall survival (OS), disease-free interval (DFI), and progression-free interval (PFI)).

Cox models were run with the "coxph" function from the "survival" package in R, and the equation for the bidirectional genes is "coxph(Surv(time,censor) ~ exprs)," where time is survival time for OS or disease-free/progression-free time for DFI/PFI, censor is survival even for OS or interval event for DFI/PFI for each patient, and exprs is the TMM normalized expression value for each gene.

For the bidirectional gene pairs, the equation is "coxph(Surv(time,censor) ~ exprs1+ exprs2)," where exprs1 and exprs2 are the TMM normalized expression value for gene1 and gene2 in a pair. We marked genes or gene pairs with Logrank $p$-value $\leq 0.05$ as prognostic genes or gene pairs.

For every prognostic bidirectional gene pair, we divided the samples into four groups based on the combination of expression level of two genes in a pair (high_vs_high, low_vs_low, high_vs_low, and low_vs_high). The cutoff to classify the groups was the median value of TMM normalized expression value for gene1 and gene2 in a pair. Then Cox models were run with the "coxph" function, and the equation for the bidirectional gene pairs is "coxph(Surv(time,censor) ~ group)." We picked the bidirectional gene pairs with Logrank $p$-value $\leq 0.05$ and considered that these bidirectional gene pairs as the pairs in which the combination of two genes with different expression level correlated with different survival outcome.

We did not perform multiple testing adjustment for survival analysis, since a previous study showed that the number of prognostic genes was no more than 50 in nine out of 16 TCGA datasets and even 0 in five out of 16 TCGA datasets after adjustment (*Anaya et al., 2015*).

## RESULTS

### Identification of bidirectional gene pairs in the human genome

We used the GENCODE gene annotation (v22) for identifying the bidirectional gene pairs present in the human genome. According to previous analyses, we defined two genes located on the opposite strands, whose TSSs were separated by less than one kb pairs, as bidirectional gene pairs (BG, head-to-head structure) (*Trinklein et al., 2004*). Meanwhile, we identified two groups of control gene pairs, which were named as control gene pairs 1 and 2 (CG1 and CG2) (Fig. 1A). Overall, we identified 4,083 bidirectional gene pairs (BG), 1,054 control gene pairs 1 (CG1), and 13,037 control gene pairs 2 (CG2). The number of bidirectional genes accounted for 12.9% of all human genes, which was consistent with the previous study (*Yang, Koehly & Elnitski, 2007*). Detailed information about all bidirectional and control gene pairs are listed in the File S1.

Among the identified bidirectional genes, 57.4% were protein coding genes. The top four gene types of bidirectional genes accounted for 89.2% of all bidirectional genes (Fig. 1B). The top four combination of gene types accounted for 75.3% of bidirectional gene pairs (Fig. 1C).

We noticed that the number and the composition of gene types for bidirectional gene pairs were different from the two control gene pairs. In order to rule out the possible bias derived from these difference, we introduced an additional control gene pairs comprising two genes randomly chosen from genome which were named as random gene pairs (Fig. 1A). The number and the composition of gene types for random gene pairs kept the same as bidirectional gene pairs (Figs. S1A and S1B). We repeated 100 times to get 100 sets of random gene pairs to avoid sampling bias.

The gene function enrichment analysis showed that terms such as DNA repair, DNA replication, and replication fork processing were enriched in bidirectional genes, accordingly with the previous study (Fig. 1D) (*Adachi & Lieber, 2002*).

### Identification and characterization of up-/down-regulated bidirectional genes in TCGA dataset

Previous studies showed that abnormal expressed genes play a crucial role in tumorigenesis and bidirectional genes may participate in the development of cancer (*Adachi & Lieber, 2002*; *Hanahan & Weinberg, 2011*; *Lee & Young, 2013*; *Yang, Koehly & Elnitski, 2007*). However, few studies systematically explored and characterized the expression profile of bidirectional genes and gene pairs in cancer. Hence, we set out to identify the differentially expressed bidirectional genes and gene pairs in 13 TCGA datasets. Detailed information of all the 13 analyzed TCGA datasets was shown in Table S1.

We used the "RUVSeq" package to correct the batch effect (*Aran et al., 2017*; *Risso et al., 2014*) and then used the "edgeR" package to identify the differentially expressed genes (*Robinson, McCarthy & Smyth, 2010*). The method we followed involved first removing genes with very low expression, which we defined as non-expressed genes. After filtration, about 50% of the all human genes in each TCGA dataset were defined as non-expressed and were removed from the subsequent analysis (Fig. S1C; File S2). Surprisingly, only about

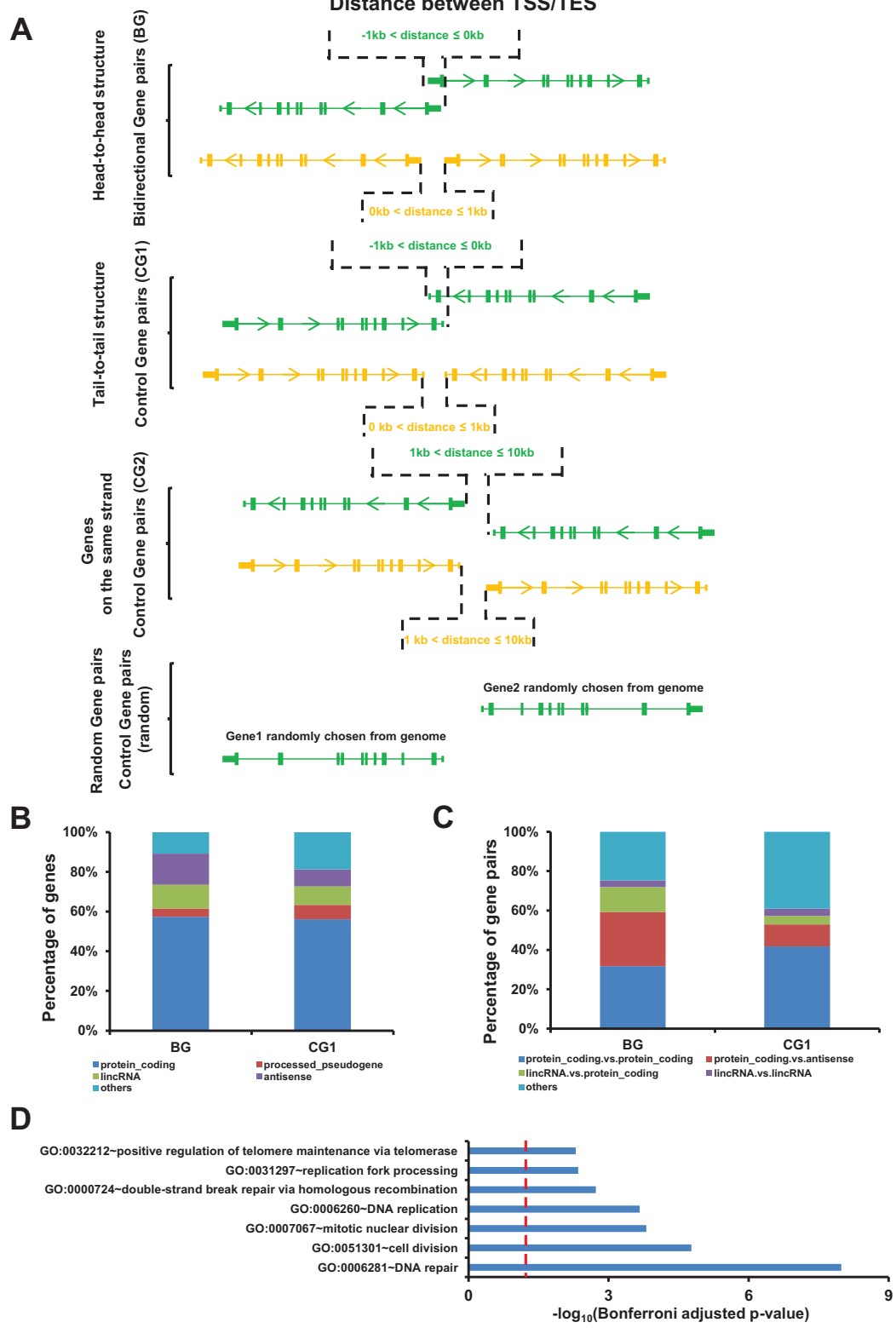

**Figure 1 Identification of bidirectional gene pairs in the human genome.** (A) Schematic representation of bidirectional gene pair (BG), control gene pair (CG1 and CG2), and random gen pair organization. TSS/TES, transcription start/end site. Arrow on gene indicating the transcriptional orientation. (B) Distribution
**Figure 1** (continued)
of gene types in bidirectional and control genes (CG1). (C) Distribution of the combinations of gene types in bidirectional and control gene pairs (CG1). (D) Gene ontology (GO) showing the function enrichment of bidirectional genes. Red dash line indicate the significant cutoff.

20% bidirectional genes in each TCGA dataset were identified as non-expressed genes (Fig. 2A). Meanwhile ~30% control and random genes were identified as non-expressed genes (Fig. 2B; Figs. S1D and S1E). The difference between the percentage of expressed genes in bidirectional genes and control/random genes in each of 13 TCGA datasets was significant ($p$-value $\leq 0.001$, "chisq.test" in R). The results indicated that bidirectional genes were prone to be expressed when comparing with control and random genes.

We then identified differentially expressed bidirectional genes by comparing the expression profiles between groups of cancer and normal samples in the 13 TCGA datasets (File S3). Although we witnessed significantly different percentage of up-/down-regulated genes between bidirectional genes and control genes in some TCGA datasets. In most of TCGA datasets there were no significant difference in the percentage of up- or down-regulated genes between bidirectional genes and control genes (Figs. 2C and 2D; Figs. S2A and S2B). We witnessed similar results by comparing percentage of up-/down-regulated genes in bidirectional genes and random genes (Figs. S3A and S3B). These results indicated that there were no consistently significant difference in percentage of up-/down-regulated genes between bidirectional and control/random genes in most of TCGA datasets.

## Identification and characterization of up-/down-regulated bidirectional gene pairs in TCGA dataset

Next, we evaluated the combination of expressed and non-expressed bidirectional genes in a pair in each TCGA dataset. Our results showed that more than 65% bidirectional genes in one pair were both expressed. Meanwhile, no more than 60% control and random genes in a pair were both expressed (Figs. 3A and 3B; Figs. S2D and S3D). The difference between the percentages of expressed gene pairs from bidirectional and control/random gene pairs were significant in all 13 TCGA datasets ($p$-value $\leq 0.001$, "chisq.test" in R). The results indicated that bidirectional genes in a pair were prone to be both expressed when comparing with control and random gene pairs.

Next, we explored the distribution of combination of up-/down-regulated bidirectional and control genes in one pair. Although the percentage of bidirectional gene pairs comprising two up- or two down-regulated genes were not always higher than the control gene pairs, we witnessed that the percentage of bidirectional gene pairs comprising two up-regulated genes plus bidirectional gene pairs comprising two down-regulated genes were significantly higher than CG1 except for LUSC dataset and CG2 except for KIRP and UCEC datasets (Figs. 3C and 3D; Fig. S2C, $p$-value $\leq 0.05$, "chisq.test" in R). We also checked the difference between the distribution of combination of up-/down-regulated bidirectional and random genes in one pair. It showed that the percentage of bidirectional

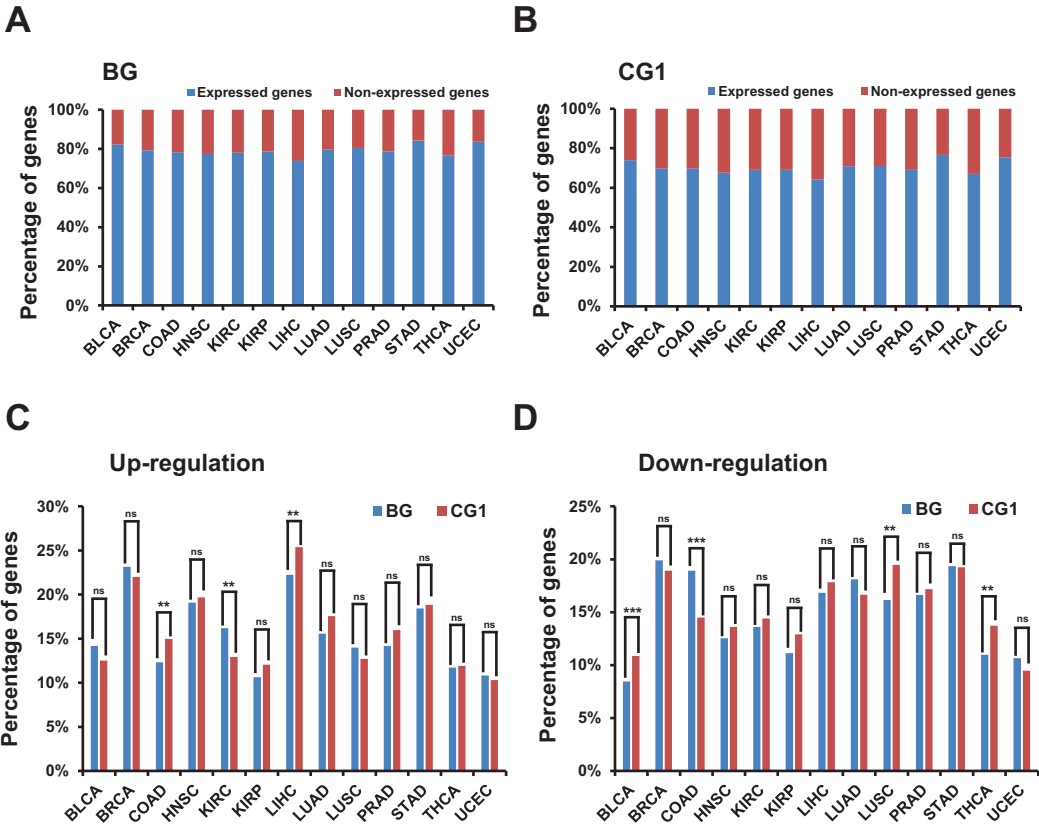

**Figure 2** **Identification and characterization of differentially expressed bidirectional genes in TCGA dataset.** (A) Percentage of expressed and non-expressed bidirectional genes among TCGA datasets. *X*-axis, TCGA datasets. (B) Percentage of expressed and non-expressed control genes (CG1) among TCGA datasets. *X*-axis, TCGA datasets. (C) Percentage of up-regulated bidirectional and control genes (CG1) among TCGA datasets. *X*-axis, TCGA datasets. The *p*-values were computed by "chisq.test" function in R. "ns" indicating not significant. "**" indicating *p*-value < 0.01. (D) Percentage of down-regulated bidirectional and control genes (CG1) among TCGA datasets. *X*-axis, TCGA datasets. The *p*-values were computed by "chisq.test" function in R. "ns" indicating not significant. "**" indicating *p*-value < 0.01. "***" indicating *p*-value < 0.001.

gene pairs comprising two up-regulated genes plus bidirectional gene pairs comprising two down-regulated genes were significantly higher than the same percentage of combination in random gene pairs in all analyzed TCGA datasets (Fig. 3C and Fig. S3C, *p*-value ≤ 0.001, "chisq.test" in R).

## Characterization of up-/down-regulated bidirectional genes and gene pairs in multiple TCGA datasets

Next, we explored up-/down-regulated bidirectional genes and gene pairs in multiple TCGA datasets. If a bidirectional gene or gene pair is up- or down-regulated in multiple cancer datasets, it indicates a common regulation and function across multiple cancer types. On the other hand, it may suggest a cancer-type specific regulation and function.

We summarized the frequency of bidirectional, control, and random genes up-/down-regulated in a specific number of TCGA datasets (Figs. 4A and 4B; File S4).
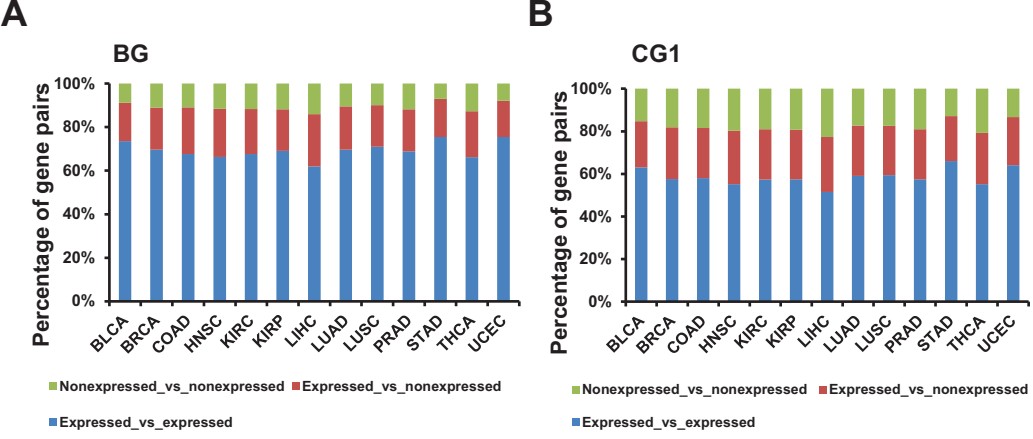

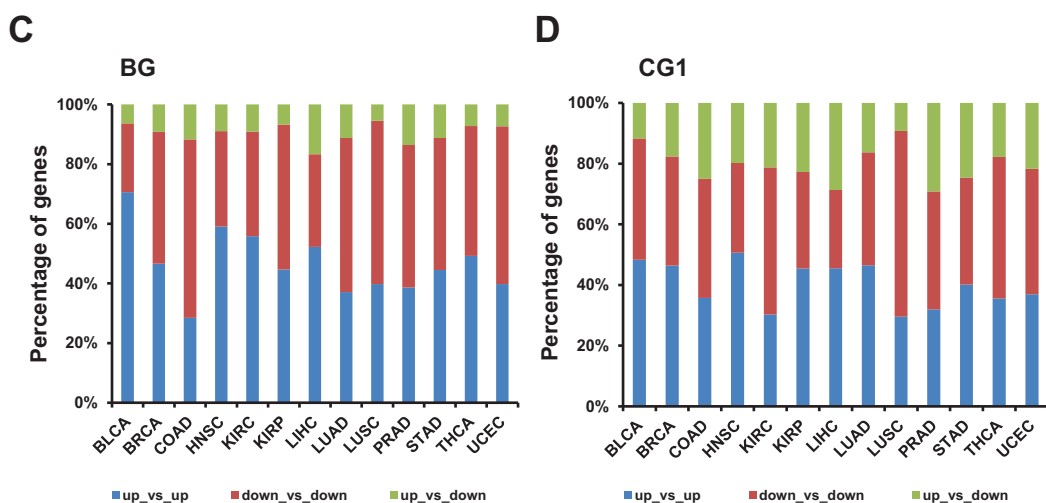

**Figure 3 Identification and characterization of differentially expressed bidirectional gene pairs in TCGA dataset.** (A) Percentage of expressed and non-expressed bidirectional gene pairs among different TCGA datasets. *X*-axis, TCGA datasets. (B) Percentage of expressed and non-expressed control gene pairs (CG1) among different TCGA datasets. *X*-axis, TCGA datasets. (C) Percentage of the patterns of differentially expressed bidirectional combination in a pair. *X*-axis, TCGA datasets. (D) Percentage of the patterns of differentially expressed control combination in a pair (CG1). *X*-axis, TCGA datasets.

Although distribution of up-regulated bidirectional genes among multiple TCGA datasets was significantly skewed to the right when comparing to the CG2 and random genes (Fig. 4A) and distribution of down-regulated bidirectional genes among multiple TCGA datasets was also significantly skewed to the right when comparing to the CG2 (Fig. 4B). The distribution of up-/down-regulated bidirectional genes showed no consistently significant pattern when comparing with the distribution of control/random genes.

Then we evaluated the distribution of bidirectional gene pairs comprising a specific combination of up-/down-regulated genes among all the 13 analyzed TCGA datasets. We observed that the distribution of bidirectional gene pairs comprising two up- or two

down-regulated genes was significantly skewed to the right when comparing to the control and random gene pairs (Fig. 4C and File S5). On the other hand, we also explored the distribution of gene pairs comprising one up- and one down-regulated genes. The result showed that the distribution of specific bidirectional gene pairs was significantly skewed to the left when comparing with CG1 or random gene pairs and no significantly different from CG2 (Fig. 4D).

Together, the results showed that the bidirectional gene pairs were prone to be up- or down-regulated in multiple TCGA datasets despite of the distribution of up- or down-regulated bidirectional genes showing no consistently significant difference from control or random genes.

## Correlation between DNA methylation and bidirectional genes

A previous study reported that CpG islands were enriched in the promoter region of bidirectional gene pairs (*Adachi & Lieber, 2002*). A recent study also witnessed the change in methylation was significantly greater in control compared to bidirectional promoters in cancer (*Thompson, Christensen & Marsit, 2018*). However, no systematical analyses have been performed for identifying and characterizing the methylation level correlated to bidirectional genes in cancer.

Here, we performed the pan-cancer analysis of methylation correlated to bidirectional genes. We defined that a methylation probe was hypo-methylated when the beta-value of this probe in cancer group was significantly lower than in the normal group, while the hyper-methylated probe when the beta-value in cancer group was significantly higher than in normal group (*Morris et al., 2014*). The detailed information of hypo-/hyper-methylated probes can be viewed in File S6. Then we defined a gene as correlated to methylation when this gene was up-regulated in cancer while at least one hypo-methylated probe was annotated to this gene or down-regulated in cancer while at least one hyper-methylated probe was annotated to this gene.

We observed no consistent trend when comparing the percentage of hypo-/hyper-methylation correlated genes between bidirectional and control genes among TCGA datasets (Figs. 5A and 5B; Figs. S4A and S4B). We also compared the percentage of hypo- or hyper-methylated bidirectional genes to the same percentage in random genes and witnessed no consistent trend (Figs. S4C and S4D).

Then we explored the combination of hyper-/hypo-methylation correlated genes in a pair in each TCGA dataset (Figs. 5C and 5D; Fig. S4E). The proportion of bidirectional gene pairs comprising two hypo-plus bidirectional gene pairs comprising two hyper-methylation correlated genes was significantly higher than the CG2 except for the BLCA dataset and than the random gene pairs in all the TCGA datasets ($p$-value $\leq 0.05$, "chisq.test" in R). We didn't compare the percentage of the combination of hyper-/hypo-methylation correlated genes in bidirectional and control gene pairs, since the number of combination of hyper-/hypo-methylation correlated genes in CG1 was limited (<5 in each combination in most of TCGA datasets) which could not produce meaningful significance.

In summary, bidirectional genes in a pair were significantly prone to be both hypo- or hyper-methylation correlated genes when comparing with the control and random gene

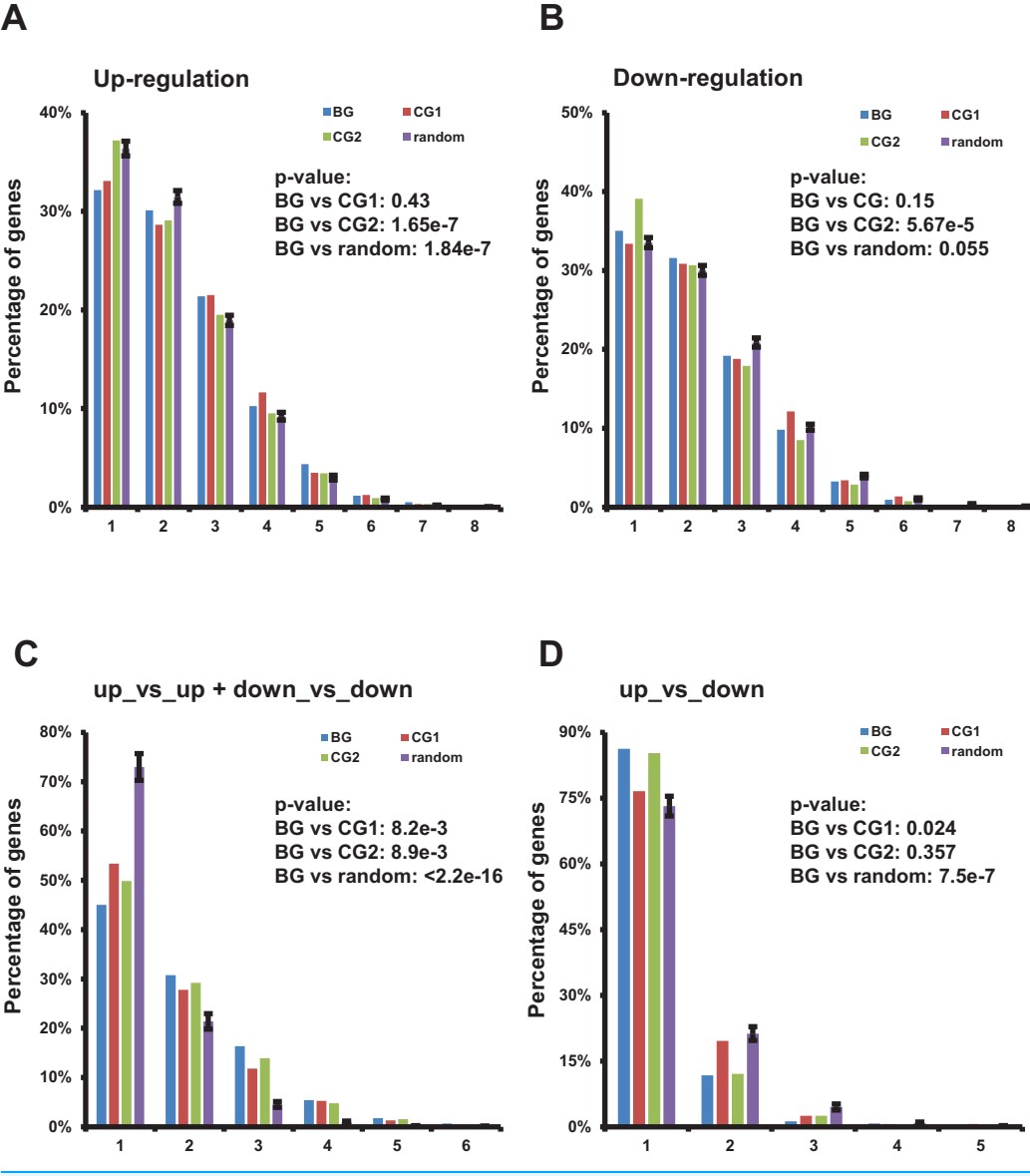

**Figure 4** **Characterization of up-/down-regulated bidirectional genes and gene pairs in multiple TCGA datasets.** (A) Percentage of bidirectional, control, and random genes up-regulated in a specific number of TCGA datasets. Each number "n" on the *X*-axis indicates the corresponding percentage of up-regulated genes on the *Y*-axis in "n" of 13 TCGA datasets. The percentages in random genes represent mean of 100 random sets and the error bars represent the standard deviation. The *p*-values were computed by "chisq.test" function in R. The number of up- or down-regulated random genes in a specific number of TCGA datasets used in "chisq.test" is the mean of 100 random sets. (B) Percentage of bidirectional, control, and random genes down-regulated in a specific number of TCGA datasets. Each number "n" on the *X*-axis indicates the corresponding percentage of down-regulated genes on the *Y*-axis in "n" of 13 TCGA datasets. The percentages in random genes represent mean of 100 random sets and the error bars represent the standard deviation. The *p*-values were computed by "chisq.test" function in R. The number of up- or down-regulated random genes in a specific number of TCGA datasets used in "chisq.test" is the mean of 100 random sets. (C) Percentage of bidirectional, control, and random gene pairs with specific combination of up- or down-regulated genes in specific number of TCGA datasets. Each number "n" on the *X*-axis means corresponding percentage on the *Y*-axis of the specific combinations of up- or down-regulated gene pairs in "n" of 13 TCGA datasets. The percentages in random genes represent mean of 100 random sets and the error bars represent the standard deviation. The *p*-values
**Figure 4** (continued)
were computed by "chisq.test" function in R. The number of specific combination in random gene pairs in a specific number of TCGA datasets used in "chisq.test" is the mean of 100 random sets. (D) Percentage of bidirectional, control, and random gene pairs with specific combination of up- or down-regulated genes in specific number of TCGA datasets. Each number "n" on the X-axis means corresponding percentage on the Y-axis of the specific combinations of up- or down-regulated gene pairs in "n" of 13 TCGA datasets. The percentages in random genes represent mean of 100 random sets and the error bars represent the standard deviation. The p-values were computed by "chisq.test" function in R. The number of specific combination in random gene pairs in a specific number of TCGA datasets used in "chisq.test" is the mean of 100 random sets.

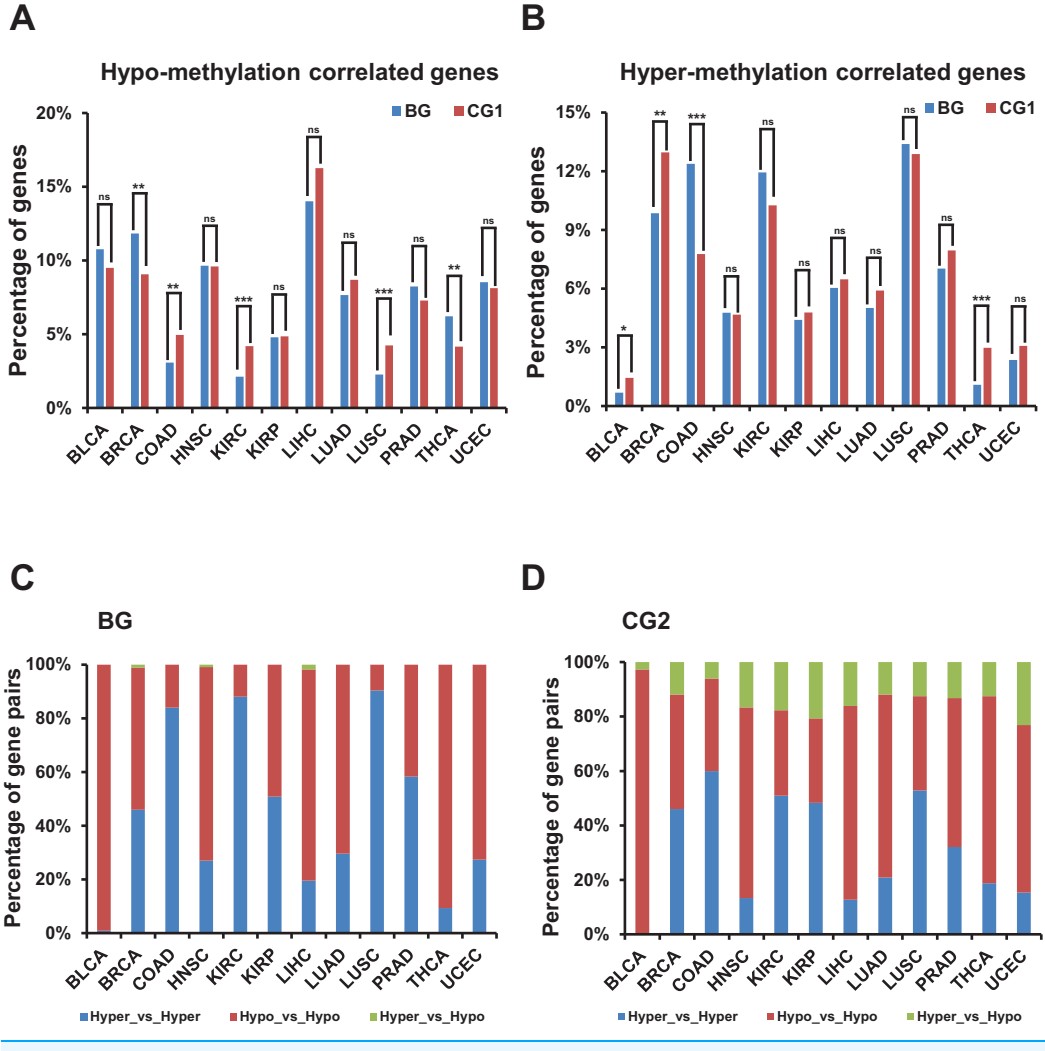

**Figure 5 Correlation of methylation levels and bidirectional genes.** (A) Percentage of hypo-methylation correlated bidirectional and control genes (CG1) in TCGA dataset. X-axis, TCGA datasets. The p-values were computed by "chisq.test" function in R. "ns" indicating not significant. "*" indicating p-value < 0.01. "***" indicating p-value < 0.001. (B) Percentage of hyper-methylation correlated bidirectional and control genes (CG1) in TCGA dataset. X-axis, TCGA datasets. The p-values were computed by "chisq.test" function in R. "ns" indicating not significant. "*" indicating p-value < 0.05. "**" indicating p-value < 0.01. "***" indicating p-value < 0.001. (C) Percentage of combination of hypo- or hyper-methylation correlated bidirectional genes in TCGA dataset. X-axis, TCGA datasets. (D) Percentage of combination of hypo- or hyper-methylation correlated control genes (CG2) in TCGA dataset. X-axis, TCGA datasets.

pairs despite of percentage of hypo-/hyper-methylation correlated bidirectional genes showing no consistently different from control and random genes.

## Correlation between prognosis and bidirectional gene pairs in TCGA datasets

In a previous study, it was indicated that bidirectional gene pairs may play a potential role in participating in the development of cancer (*Yang, Koehly & Elnitski, 2007*). A previous study also showed that certain transcriptional factor can bind and repress the transcription of BRCA1 from its bidirectional promoter in cancer (*Di et al., 2010*). In order to explore the potential clinical outcome of bidirectional genes, we identified prognostic bidirectional and control genes and gene pairs by survival analysis for three endpoints (OS, DFI, and PFI). We marked genes or gene pairs with Logrank $p$-value $\leq 0.05$ as prognostic genes or gene pairs. The results of survival analysis for all three endpoints in BG and CG1/2 can be viewed in Files S7–S9.

Since the protein coding genes are the major function executor compared to other gene types in cells. We firstly evaluated the percentage of prognostic bidirectional genes from protein coding genes. However, there was no consistent trend that the percentage of prognostic protein coding genes in bidirectional genes were significantly higher or lower than control genes in survival analysis for OS, DFI, and PFI (Table S2). There were also no significant and consistent difference from bidirectional and control genes from all human genes in the percentage of prognostic genes in survival analysis for OS, DFI, and PFI (Table S3).

Then we checked the distribution of prognostic bidirectional gene pairs in 13 TCGA datasets (Files S10–S12). Briefly, there were no significantly different between bidirectional and control gene pairs in percentage of prognostic gene pairs for survival analysis for all three-time endpoints in most of TCGA datasets. (Tables S4 and S5) and no consistent trend in both gene pairs comprising two protein coding genes and all genes. However, we noticed that the percentage of prognostic bidirectional genes/pairs in LIHC dataset was significantly and consistently higher than the corresponding percentage in control genes/pairs in all analysis except for the gene pairs in survival analysis for DFI.

Remarkably, we found that the combination of two genes with different expression level correlated with different survival prognosis in bidirectional and control gene pairs among prognostic gene pairs. The percentage of these bidirectional gene pairs comprising two protein coding genes were significantly higher than control gene pairs (CG1 and CG2) in COAD dataset and significantly lower in none of 13 TCGA datasets in survival analysis for OS (Figs. 6A and 6B; File S13).

For example, the two bidirectional genes ENSG00000040531.13 and ENSG00000262304.1 forming a gene pair, were both prognostic genes in survival analysis for OS, and the down-regulation of both genes may shorten the survival time in HNSC dataset (Figs. 6C and 6D). Moreover, low expression of both genes was associated with the poorest prognosis compared with high expression of one gene and high expression of both genes which was associated with the best prognosis in survival analysis for OS (Fig. 6E).

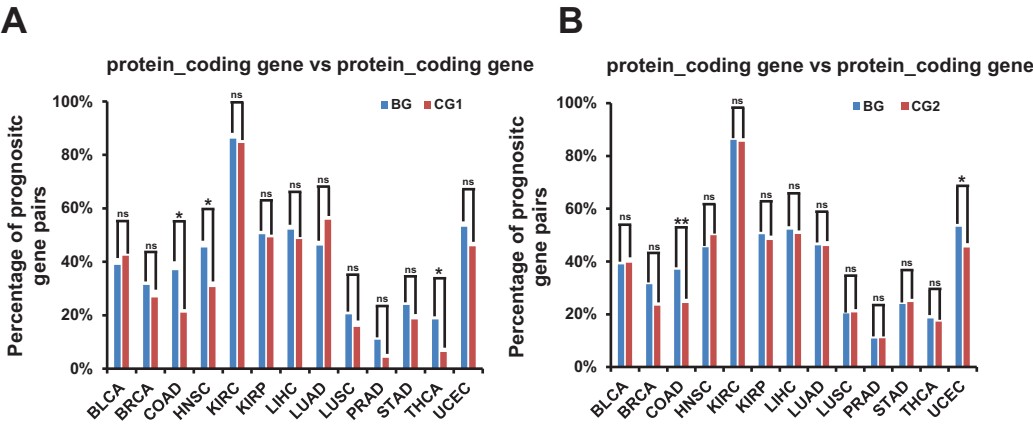

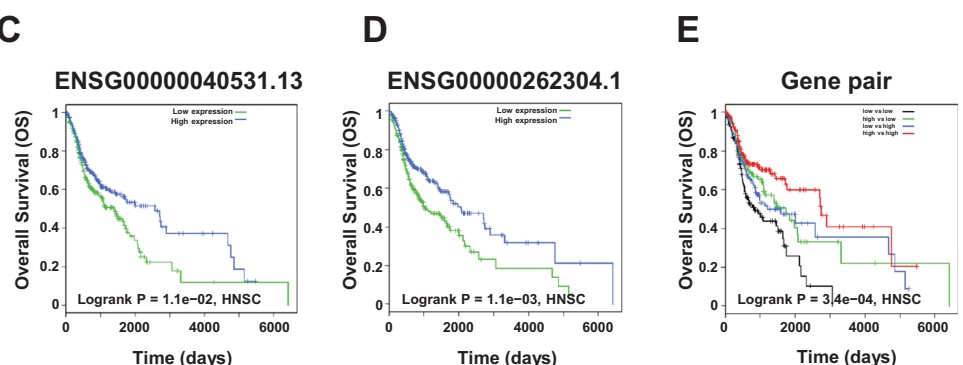

**Figure 6 Correlation of survival rate and bidirectional genes or gene pairs in TCGA datasets.**
(A) Percentage of candidate prognostic gene pairs comprising two protein coding genes among all prognostic bidirectional and control gene pairs (CG1) comprising two protein coding genes in survival analysis for OS. Candidate prognostic gene pair were identified as two genes in this pair with different expression level correlated with different survival outcome. The *p*-values were computed by "chisq.test" function in R. "ns" indicating not significant. "*" indicating *p*-value < 0.05. (B) Percentage of candidate prognostic gene pairs comprising two protein coding genes among all prognostic bidirectional and control gene pairs (CG2) comprising two protein coding genes in survival analysis for OS. Candidate prognostic gene pair were identified as two genes in this pair with different expression level correlated with different survival outcome. The *p*-values were computed by "chisq.test" function in R. "ns" indicating not significant. "*" indicating *p*-value < 0.05. "**" indicating *p*-value < 0.01. (C) Survival plot for bidirectional genes, ENSG00000040531.13. Median value of TMM normalized expression value was used as cutoff to divide samples into four groups. (D) Survival plot for bidirectional genes, ENSG00000262304.1. Median value of TMM normalized expression value was used as cutoff to divide samples into four groups. (E) Survival plot for bidirectional gene pair. Median value of TMM normalized expression value was used as cutoff to divide samples into four groups.

## DISCUSSION

By analyzing the gene expression profiles in TCGA datasets, we identified the up- and down-regulated bidirectional genes and gene pairs. It provided an overview of expression pattern of bidirectional genes and shed a light on the function of bidirectional genes in cancer. In a previous study, it indicated that the genes in a bidirectional gene pair were prone

to be positively co-expressed (*Trinklein et al., 2004*). The underlying mechanism was considered to be controlled by a common promoter (bidirectional promoter) and be the co-regulation of two genes in a pair. Our results indicated that the percentage of up-/down-regulated bidirectional genes showed no significant difference from the control and random genes in most of TCGA datasets. However, the percentage of bidirectional gene pairs comprising two up-regulated genes plus gene pairs comprising two down-regulated genes was significantly higher than the control and random gene pairs in almost of all TCGA datasets. Bidirectional promoter and co-regulation of two genes in a pair may contribute to this phenomenon. It also indicated that the emphasis on the regulation of bidirectional genes lied on the regulation in the pair manner. Besides, a small portion of bidirectional gene pairs comprised one up-regulated gene and one down-regulated gene which was consistent with co-expression analysis from a previous study (*Trinklein et al., 2004*). Although it can be explained by post-transcription process, such as regulation by microRNA, or different regulation mechanisms on transcription level underlying two genes in a pair. Further studies should be needed to clarify these hypotheses.

We also identified the methylation correlated bidirectional genes. Identification of these methylation correlated genes will be useful for further study in regulation mechanism of bidirectional gene pairs in different cancer types. Like the differentially expressed gene pairs, the bidirectional gene pairs were also prone to be both hypo- or hyper-methylation correlated genes when comparing to the random gene pairs. It suggested that differential methylation may contribute to the expression of bidirectional genes. Again, the emphasis on the regulation of methylation annotated to the bidirectional genes was in the pair manner.

Finally, we found that the combination of two genes in a pair with different expression level correlated with different survival outcome for OS. Although the percentage of these gene pairs in bidirectional gene pair were significantly higher than in control gene pair in only one TCGA datasets. It still may indicate functional overlapping of two genes in a bidirectional gene pair.

## CONCLUSIONS

In this study, we found that the bidirectional genes in a pair were prone to be both up-/down-regulated and regulated by both hypo-/hyper-methylation. Our results emphasized the unique role of expression and regulation of bidirectional genes in pair manner and the functional consequences of both genes in a pair should be considered in the future study.

## ACKNOWLEDGEMENTS

We acknowledge the contributions of the TCGA Research Network. Without their efforts this type of analysis would not be possible.

### Funding

The authors received no funding for this work.

## Competing Interests

The authors declare that they have no competing interests.

## Author Contributions

- Juchuanli Tu conceived and designed the experiments, performed the experiments, analyzed the data, contributed reagents/materials/analysis tools, prepared figures and/or tables, authored or reviewed drafts of the paper, approved the final draft.
- Xiaolu Li authored or reviewed drafts of the paper, approved the final draft.
- Jianjun Wang conceived and designed the experiments, prepared figures and/or tables, authored or reviewed drafts of the paper, approved the final draft.

## Data Availability

The raw measurements are provided in the Supplemental Files.

## Supplemental Information

Supplemental information for this article can be found online at http://dx.doi.org/10.7717/peerj.7107#supplemental-information.

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
