# Peer review of "Characterization of bidirectional gene pairs in The Cancer Genome Atlas (TCGA) dataset"

_PeerJ, doi:10.7717/peerj.7107_

## Round 0.1 · original submission · Minor Revisions

The reviewers have raised a few concerns regarding the usage and interpretation of TCGA data, including methylation status of samples, clinical covariates, and correction of batch effects. I believe that addressing these comments will improve the revised manuscript. Also I recommend further editing of the paper to correct grammatical mistakes and typos.

New:

Reviewer 1 ·

Basic reporting

In the abstract, the authors should clearly specify the controls used. For example: (i) adding a sentence like "Gene pair on the opposite strand with transcription end site distance within 1 kb (tail-to-tail structure) were used as control", (ii) adding a phrase like "in tumor compared to adjacent normal" instead of just saying "up/down regulated" or "hypo/hyper methylated", etc. While the authors have already specified these in the "Materials and Methods" section, readers often look at the abstract separately, and omission of these crucial details can make the abstract difficult to understand.

Experimental design

Satisfactory

Validity of the findings

1. In the methylation analysis, the authors used all the CpG probes annotated to a gene, i.e. the CpG probes located in the promoter region of a gene as well as the CpG probes located elsewhere in the gene. While hypo/hyper methylation of the former type of probes is easier to interpret in the context of expression, it is not always clear how to interpret hypo/hyper methylation of the later type of probes in the context of expression. So, the authors should check whether restricting the analysis to only the CpG probes located in the promoter region (say 1 kb up-stream) of a gene, i.e. excluding the CpG probes located elsewhere in the gene from analysis, produces results that are consistent with their current findings (and add a few sentences and possibly a supplementary figure on this).

2. TCGA has recently published a well-curated version of their pan-cancer clinical data (please see the tab "TCGA-CDR" in Table S1 of Liu et al, Cell 2018, PubMed 29625055), detailing not only overall survival (OS) but also disease-specific survival (DSS), disease-free interval (DFI), and progression-free interval (PFI). The authors should check whether analysis of these data produces results that are consistent with their current findings (and add a few sentences and possibly supplementary figures on this). These supporting results can considerably strengthen the claims made by the authors.

Additional comments

The result section "Characterization of up-/down-regulated bidirectional genes and gene pairs in multiple TCGA datasets" (line 287-319) incorrectly refers to Figure 3 (should refer to Figure 4 instead).

Reviewer 2 ·

Basic reporting

This manuscript is very well presented in language and provides thorough research background and well performed analysis result. Some crucial typos need to be revised before acceptance:
1. lines 294-lines 312, the figure mentioned should always be Fig.4
2. line 409, Fig.6E should be Fig.6D

Experimental design

How did the authors deal with batch effect when performing differential expression analysis?

Validity of the findings

The authors had a list of interesting findings when comparing bidirectional genes and control genes, especially the identified correlation with prognosis. However, the result interpretation is not thorough enough. For gene set enrichment, DAVID GO analysis is far from enough. I would suggest the authors to try extra tools for interpretation like Enrichr (Kuleshov et al., 2016), which keeps a updated database of GO, pathways and etc.

Additional comments

The paper is well written. The finding is quite interesting. Better result interpretation and some validation experiments will elevate this paper a lot.

·

Basic reporting

No Comment.

Experimental design

The analyses needs additional comparison with controls, or the authors might want to give a satisfactory explanation for why the control chosen by them proves to be sufficient in this study.

Line 109: Do the authors include both “blood” and “solid tissue” within the normal group? If so then only “solid tissue” normal must be considered as the RNA profile of a blood sample may vary greatly from that of normal tissue and hence a differential expression compared to the tumor may not be a sensitive predictor.survi

Line 120: When filtering for logFold how were the cut-offs +1.5 and -0.67 defined? Xiao et al 2018, does not give a reason for different cut-offs in the upper and lower limits for log fold change. I believe a different fold change cut-off might end up skewing the results. Further Xiao et al used multtest R package to compute FDR as against glmTreat native edgeR function by the authors. Are the authors confident that the two different packages predict similar FDR values?

Line 136: Similar concern for whether “blood normals” were also used in the normal group to identify differentially methylated genes.


Line 174: The number of controls is lower than the test set. What is the reason on tail to tail gene pairs are used as controls? For the control pairs, additional controls, like pair of genes in the same strand where the start of one gene is >1kb and <5kb from the stop of the other, can be used as such pairs might show co-transcriptional pattern. Further shuffling the location of all genes on the genome and then re-identifying the bidirectional pairs, is another random control which needs to be included in the study.

Validity of the findings

Results section is not often accompanied by a sound conclusion. However the nature of the study is as such that often a direct biological inference is difficult to obtain. Hence would recommend the authors to tone down their conclusion and omit or reduce statistics which do not have a clear biological hypothesis, thus making the message in the paper more clear and concise.

Line 263-273: Here the authors make certain observations, however there is a lack of inference, and whether the observations reflect the biology of cancer genomes, or the gene orientations in general.


Line 301: This is not necessary a cancer specific observation, but more prominent due to tissue specificity for Head to Head gene pairs, as also shown before in Fig 6 of a previously published paper (https://www.ncbi.nlm.nih.gov/pmc/articles/PMC2665707/). Hence I think the authors must be careful in their interpretation since, and use additional control sets to get a clear picture.


Line 311: Did not find Fig. 3 C, D and would recommend to refer to the previous publication
https://www.ncbi.nlm.nih.gov/pmc/articles/PMC2665707/
Which shows similar findings for Head to Head gene pairs.

Line 356-357: Here the text lacks any biological inference from the observation, hence it is not clear as to the purpose of the statistics. I would recommend to make concise all statistical observations (or even omit), which do not provide a clear biological hypothesis as it makes the core message in the manuscript a bit obscure.

Line 371-409: Bidirectional genes compared to any other category, tend to be significantly enriched in genes important for chromatin state, replication and repair. Hence their relation to prognosis might be reflective of their underlying function, and change in expression due to tumor state or treatment, and not because of the genes being important in that particular type of cancer. Hence the any conclusion drawn from such an analysis, needs to be toned down.

Additional comments

The current study looks at the role of bidirectional genes in head to head orientation with respect to cancer and gene expression and attempts to elucidate the expression patterns and functional role of such pairs in cancer. The study is well planned and technically sound and would be interesting for the community. However I strongly believe the authors must include or at least test their results with additional controls (same stranded gene pairs and random pairs) and get a more holistic picture as well as reduce the text about those statistics which do not provide a clear biological inference. Also an aspect of concern would be if the normals used in the study include blood normals, since in TCGA RNAseq the blood normals are not a good measure of normal tissue expression of a gene (https://tcga-data.nci.nih.gov/docs/publications/tcga/about.html). However the manuscript is fit for publication considering the authors address the issues and also include additional controls or give a satisfying explanation to why only tail to tail pairs were used as controls.

---

## Round 0.2 · Minor Revisions

I am pleased to inform you that it is potentially acceptable for publication, once you have carried out some minor revisions suggested by our reviewers, particularly, in presentation of the data and results.

Reviewer 1 ·

Basic reporting

The authors have satisfactorily implemented the changes I recommended in my last review.

Experimental design

no comment

Validity of the findings

The authors have satisfactorily implemented the changes I recommended in my last review.

·

Basic reporting

Maybe add a supplementary text document with additional figures and text, thus cutting down on statistical information which is needed but tends to hide major points of the manuscript in the main text.

The Figures, specially in supplementary need minor corrections. Overall the structure of the manuscript is still slightly confusing with lot of information, which is at times too much for a reader, without deriving a clear stock message.

Minor things:
Figure S1A should be a table

Figure S3,S4, S6 not marked in A and B

Figure S5 E,F,G,H unavailable, or A, B, C, D incorrectly labelled

Experimental design

Strongly suggest to add as control gene pairs which are transcribed in the same direction and separated by 1 Kb to 10 Kb. It is important to show this group as a control along with tail to tail, and random pairs, as similar to the test dataset (head to head), genes transcribed in the same strand and lying close to each other, may often fall under a common transcriptional domain, thus showing a similar behavior as head to head group when it comes to differential normal/tumor expression as well as gene function and survival analysis.

Validity of the findings

The study adds to the field and is thus worth publishing. However the results need to be trimmed and explained in a concise manner without overbearing the reader (many could be biologists) with statistics.

Additional comments

The manuscript is descriptive and lot of information remains hidden unless carefully observed by the reader. Hence making it more concise, highlighting the principal findings with figures in the main text, and keeping the rest of the details, which is of course hard work of authors, in a supplementary text file would be suitable. Would recommend adding a control set of gene pairs transcribed in the same strand and lying close to each other (1-10 kb).

---

## Round 0.3 · accepted · Accept

This is a comprehensive and well-controlled analysis of TCGA data. Congratulations on the acceptance of your paper!